# Sodium Nitrite Attenuates Reduced Activity of Vascular Matrix Metalloproteinase-2 and Vascular Hyper-Reactivity and Increased Systolic Blood Pressure Induced by the Placental Ischemia Model of Preeclampsia in Anesthetized Rats

**DOI:** 10.3390/ijms241612818

**Published:** 2023-08-15

**Authors:** Laisla Zanetoni Martins, Maria Luiza Santos da Silva, Serginara David Rodrigues, Sáskia Estela Biasotti Gomes, Laura Molezini, Elen Rizzi, Marcelo Freitas Montenegro, Carlos Alan Dias-Junior

**Affiliations:** 1Department of Biophysics and Pharmacology, Institute of Biosciences, Sao Paulo State University (UNESP), Botucatu 18618-689, Brazil; laisla.martins@unesp.br (L.Z.M.); mls.silva@unesp.br (M.L.S.d.S.); serginara.rodrigues@unesp.br (S.D.R.); saskia.estela@unesp.br (S.E.B.G.); 2Unit of Biotechnology, University of Ribeirao Preto (UNAERP), Ribeirao Preto 14096-900, Brazil; laura.molezini@sou.unaerp.edu.br (L.M.); ersanchez@unaerp.br (E.R.); 3Department of Molecular Biosciences, The Wenner-Gren Institute, Stockholm University, 10691 Stockholm, Sweden; marcelo.montenegro@su.se

**Keywords:** preeclampsia, metalloproteinase-2, nitric oxide, sodium nitrite, placental ischemia

## Abstract

Preeclampsia is a maternal hypertension disorder associated with vascular dysfunction and fetal and placental growth restrictions. Placental ischemia is suggested as the primary trigger of preeclampsia-associated impairments of both endothelium-derived nitric oxide (NO) and the vascular activity of extracellular matrix metalloproteinase-2 (MMP-2). Reduced uteroplacental perfusion pressure (RUPP) is a placental ischemia model of preeclampsia. Reduction of sodium nitrite to NO may occur during ischemic conditions. However, sodium nitrite effects in the RUPP model of preeclampsia have not yet been investigated. Pregnant rats were divided into four groups: normotensive pregnant rats (Norm-Preg), pregnant rats treated with sodium nitrite (Preg + Nitrite), preeclamptic rats (RUPP), and preeclamptic rats treated with sodium nitrite (RUPP + Nitrite). Maternal blood pressure and fetal and placental parameters were recorded. Vascular function, circulating NO metabolites, and the gelatinolytic activity of vascular MMP-2 were also examined. Sodium nitrite attenuates increased blood pressure, prevents fetal and placental weight loss, counteracts vascular hyper-reactivity, and partially restores NO metabolites and MMP-2 activity. In conclusion, sodium nitrite reduction to NO may occur during RUPP-induced placental ischemia, thereby attenuating increased blood pressure, fetal and placental growth restriction, and vascular hyper-reactivity associated with preeclampsia and possibly restoring NO and MMP-2 activity, which underlie the blood pressure-lowering effects.

## 1. Introduction

Preeclampsia is a hypertensive disorder of pregnancy that affects about 3–8% of women worldwide [1]. Preeclampsia is characterized by maternal hypertension that may disturb pregnant women’s and fetal health [2]. The pathogenesis of preeclampsia is still poorly understood [2]. One of the main theories related to the causes of preeclampsia is a deficient trophoblastic invasion of the maternal endometrium, which results in insufficient placentation, thus reducing uteroplacental blood flow with consequent placental ischemia and fetal hypoxia [2]. Preeclamptic ischemic placenta releases various bioactive factors in the maternal circulation that are associated with endothelial dysfunction and compromise the action of endogenous NO [3,4,5].

NO is a vasodilator molecule produced by vascular endothelial cells, controls maternal blood pressure in pregnancy, and regulates uteroplacental blood flow [6,7]. Placental development and maturation are complex processes that require coordinated regulation of trophoblast invasion, differentiation, and proliferation in the maternal decidua [8]. Importantly, trophoblast differentiation, motility, and invasion may all be dysregulated in preeclampsia [9]. Moreover, dysfunction of endogenous NO production in the maternal circulation may also contribute to pregnancy complications, including preeclampsia [9].

In the recent past, the NO metabolite, named nitrite, was considered an inert product of NO metabolism [10,11]. However, although endothelium-derived NO is the classic pathway for endogenous NO synthesis, recent research suggests that nitrite is an alternative pathway for NO formation in the human organism [12]. Indeed, nitrite has been considered a bioactive storage reservoir for NO, and reducing nitrite to NO has been considered a pharmacological strategy to restore circulating NO levels [13]. Interestingly, the nitrite reduction to NO seems to occur mainly in conditions of hypoxia, acidosis, and ischemia [14,15,16,17,18]. As demonstrated in a previous study in vitro, inorganic nitrite vasodilated human placental arteries obtained after cesarean sections following uncomplicated pregnancies [15].

The intentional increase in NO bioavailability after treatment with inorganic nitrite has also been shown to attenuate endothelial dysfunction and arterial hypertension [19,20,21]. Previous experimental studies provide evidence that antihypertensive effects related to nitrite treatment may also involve the improvement of endothelial function [19,20], including in a model of pregnancy hypertension induced by NO synthase inhibition in rats [6]. However, the effects of sodium nitrite treatment are still unclear on placental ischemia-induced preeclampsia, which is an important event in the pathogenesis of preeclampsia.

It has been experimentally demonstrated that the endothelium-derived NO-induced vasodilation impaired by preeclampsia is also associated with increases in vascular contractions [5,22] and that this preeclampsia-induced vascular dysfunction may also be related to decreases in the proteolytic activity of important proteases involved in the vascular remodeling, such as extracellular matrix metalloproteinase-2 (MMP-2) [23,24,25,26]. Indeed, MMP-2 is a gelatinase that degrades various components of the extracellular matrix, such as collagen and elastin found in the walls of blood vessels [27]. The pregnancy-associated changes involve significant uteroplacental and vascular remodeling, which are, at least in part, regulated by MMP-2 [28]. Hence, the balanced activity of MMP-2 may be necessary for maintaining normal vascular tone, including in normotensive pregnancy [28], while the dysregulated activity of MMP-2 has been associated with vascular dysfunction in pregnancy hypertension [9,23,24,25,26,28]. Accordingly, decreases in the gelatinolytic activity of vascular MMP-2 [23,25,29] associated with reduced NO metabolites in the maternal circulation [5,30] and the development of hypertension in preeclamptic pregnancies have also been observed. However, no previous study has yet examined the effects of sodium nitrite in the RUPP model of preeclampsia. Therefore, in the present study, we hypothesized that RUPP-induced placental ischemia promotes nitrite reduction to NO, thus restoring the circulating NO bioavailability that protects against pregnancy hypertension and vascular dysfunction associated with the reduced gelatinolytic activity of vascular MMP-2.

## 2. Results

### 2.1. Sodium Nitrite Effects on Systolic Blood Pressure and Fetal and Placental Parameters

As previously mentioned, the induction of pre-eclamptic syndrome occurred on the 14th day of pregnancy. Systolic blood pressure (SBP) values were not different among groups on the day before RUPP surgery.

Significant increases in maternal SBP were observed in the RUPP group when compared to the Norm-Preg group. On the other hand, sodium nitrite treatment attenuated the increase in SBP caused by RUPP in the RUPP + Nitrite group (Figure 1 and Appendix A). In addition, the RUPP + Nitrite group still demonstrated a significant increase in SBP in comparison with the Preg + Nitrite group. We have also observed that the SBP in the Preg + Nitrite group was lower in relation to those found in the Norm-Preg group, indicating a hypotensive effect of sodium nitrite (Preg + Nitrite group, Figure 1 and Appendix A).

Fetal weight was reduced in RUPP rats when compared to pregnant rats treated (or not) with sodium nitrite (Figure 2A and Appendix A). The number of pups (Litter size) was reduced in RUPP rats when compared to the Norm-Preg group (Figure 2B and Appendix A). Placental weight was also reduced in RUPP rats when compared to pregnant rats treated (or not) with sodium nitrite (Figure 2C and Appendix A). The RUPP + Nitrite group presented restoration of fetal weight, litter size, and placental weight with significant increases when compared to the RUPP group (Figure 2A–C). In addition, fetal weight in the RUPP + Nitrite group is also different from the Preg + Nitrite group (Figure 2A and Appendix A). As previously observed in SBP, nitrite had a hypotensive effect and also reduced fetal weight and litter size in pregnant treated rats (Preg + Nitrite group, Figure 2A,B and Appendix A), thereby suggesting that nitrite-derived NO ameliorated the blood supply to the growing fetus in the RUPP + Nitrite group, while nitrite-derived NO has likely caused maternal hypotension, which leads to a reduction in the maternal-fetal interface blood flow, thus impairing fetal growth in the Preg + Nitrite group.

### 2.2. Sodium Nitrite Treatment and Circulating and Placental Levels of NO Metabolites

Significant reductions in circulating NO metabolites in plasma were found in the RUPP group compared to the Norm-Preg group. In contrast, the RUPP + Nitrite group showed no statistical differences related to the Norm-Preg and RUPP groups, indicating that intermediate values (between the Norm-Preg and RUPP groups) regarding NO metabolites have been reached in the RUPP + Nitrite rats (Figure 3 and Appendix A). On the other hand, a significant difference was observed between RUPP + Nitrite compared to the Preg + Nitrite group (Figure 3 and Appendix A).

Regarding placental NO metabolites, significant increases were found in animals from the RUPP and RUPP + Nitrite groups compared with sham-operated animals treated (or not) with sodium nitrite (Norm-Preg and Preg + Nitrite groups; Figure 4 and Appendix A). The RUPP + Nitrite group showed no significant difference in relation to the RUPP group (Figure 4 and Appendix A), thereby indicating that sodium nitrite treatment produced no effects on uteroplacental ischemia-induced increases in placental NO metabolites.

### 2.3. Sodium Nitrite Effects on Vascular Hyperreactivity and Endothelial Dysfunction

Aortic rings from RUPP animals were shown to be more reactive to phenylephrine-induced vasoconstriction compared to the normotensive rats (Norm-Preg group; Figure 5A and Appendix A), and this difference was even more evident when the endothelium was removed (Figure 5B and Appendix A).

Sodium nitrite attenuated phenylephrine-induced vasoconstriction in the RUPP + Nitrite group when compared to those found in the RUPP group (Figure 5A,B and Appendix A). However, the last point of the concentration-response curve did not show this difference in endothelium-intact aortic rings (RUPP + Nitrite vs. RUPP group, Figure 5A and Appendix A), while endothelium-denuded aorta showed a significant difference on the last point of the concentration-response curve (Figure 5B and Appendix A).

Acetylcholine-induced vasodilation in aortic rings with intact endothelium was impaired in both RUPP and RUPP + Nitrite groups in comparison with untreated and treated sham-operated rats (Norm-Preg and Preg + Nitrite groups, Figure 6A and Appendix A).

In endothelium-denuded aortic rings, there was a complete absence of acetylcholine-induced relaxation (Figure 6B and Appendix A). Moreover, in endothelium-intact aortas incubated with L-NAME (a NO synthase inhibitor), there was total blockage of acetylcholine-induced vasodilation (Figure 6C and Appendix A). Thus, mechanical endothelium removal or pharmacological blockade of NO synthase with L-NAME provides further evidence that endothelium-derived NO is involved and is impaired in endothelium dysfunction induced by uteroplacental ischemia.

### 2.4. Sodium Nitrite Effects on the Gelatinolytic Activity of Vascular MMP-2

Zymography analysis of the abdominal aorta tissue homogenates revealed proteolytic bands corresponding to the 72 kDa and 64 kDa MMP-2 isoforms (Appendix A), while the bands corresponding to the 75 kDa (MMP-2) and 92 kDa (MMP-9) isoforms were not detected (Appendix A).

The gelatinolytic activity of the inactive gelatinase (72 kDa MMP-2) was higher in the Norm-Preg and Preg + Nitrite groups when compared to the RUPP and RUPP + Nitrite groups (Figure 7A and Appendix A). Additionally, there was a greater gelatinolytic activity of the inactive MMP-2 isoform (72 kDa) in the RUPP + Nitrite group compared to the RUPP group (Figure 7A and Appendix A). Furthermore, the gelatinolytic activity of the active MMP-2 isoform (64 kDa) was also higher in the Norm-Preg and Preg + Nitrite groups compared to the RUPP and RUPP + Nitrite groups (Figure 7B and Appendix A). Moreover, the RUPP + Nitrite group exhibited higher gelatinolytic activity of the active MMP-2 isoform (64 kDa) than the RUPP group (Figure 7B and Appendix A).

## 3. Discussion

The present findings demonstrate, for the first time, that sodium nitrite attenuates the increases in systolic blood pressure, counteracts the vascular hyper-reactivity, and prevents fetal and placental weight loss induced by the RUPP model of preeclampsia. This study also shows that sodium nitrite partially restores the gelatinolytic activity of vascular MMP-2 impaired by RUPP. Thus, our results suggest that reduction of nitrite to NO occurs during placental ischemia and endothelial dysfunction, thereby protecting against RUPP-associated increases in vascular contraction and in systolic blood pressure.

Reduced vascular MMP-2 gelatinase activity and vascular hyper-reactivity have been observed in models of preeclampsia induced by uteroplacental ischemia [5,31,32], which is caused by the reduction of uterine perfusion pressure (RUPP) in rats. Thus, decreased MMP-2 activity may also contribute to the pathophysiological vascular remodeling during preeclampsia, as previously suggested [23,25]. Indeed, an important event in the pathogenesis of preeclampsia is the inadequate endovascular uterine invasion of trophoblasts, which leads to the failure of spiral artery transformation and thereby results in an insufficient blood supply to the uteroplacental interface with consequent placental ischemia [2]. Ischemic placental release of cytotoxic factors into the maternal circulation has been suggested as a trigger in the pathogenesis of preeclampsia [4,5,33]. There is also previous evidence that this initial event could lead to the development of impaired endothelial function associated with decreased MMP-2 activity in the maternal vasculature of preeclamptic rats [32]. Accordingly, the present results show increases in systolic blood pressure and decreases in fetal and placental weights, which may be associated with endothelial dysfunction and reduced MMP-2 activity in RUPP compared with the Norm-Preg group. We have also assessed the extent of vascular dysfunction, and increases in vascular contraction to the alpha-adrenergic receptor agonist (phenylephrine) in aortas from systemic circulation were found, which are corroborated by earlier studies in preeclamptic models [5,31,32] and women with preeclampsia [34,35].

The effects of sodium nitrite treatment on blood pressure and vascular function were also examined. We observed that the increases in systolic blood pressure, weight loss of fetuses and placentas, and hyper-reactivity induced by phenylephrine were all attenuated with sodium nitrite administration in the RUPP + Nitrite rats. The blood pressure-lowering effects, protection of fetal and placental development, and prevention of the increases in phenylephrine-induced vascular contractions in RUPP + Nitrite rats could have involved sodium nitrite-derived NO since increases in circulating NO metabolites in the plasma of the Preg + Nitrite group were found. However, reductions in NO metabolites in plasma were observed in the RUPP compared with the Norm-Preg group. Moreover, intermediate concentrations (between the Norm-Preg and RUPP groups) of NO metabolites were detected in plasma from RUPP + Nitrite rats, and they were not statistically different from the Norm-Preg group (Figure 3 and Appendix A). Thus, the present results suggest that nitrite-derived NO in maternal circulation counteracted the vascular hyper-reactivity and restored blood flow to supply fetal and placental growth in the RUPP + Nitrite group (Figure 3 and Appendix A).

Recent studies have suggested that sodium nitrite-derived NO orally administered could attenuate arterial hypertension in animals [12,19,20,36,37,38] and humans [39], including pregnancy hypertension [6,7]. It has also been demonstrated that ionized nitrite is recycled back to NO, thus providing an alternative to restore NO levels independently of NO synthesis by the endothelium [12,40]. Importantly, endogenous NO formation from endothelium is impaired in preeclamptic RUPP rats [5,41] and in women with preeclampsia [42,43]. The present study demonstrates that endothelial dysfunction featured by impaired acetylcholine-induced vasodilation was slightly improved in the RUPP + Nitrite group because no differences were found in the respective concentration-induced relaxations of the Acetylcholine (10^−6^ mol/L, Figure 6A) in comparison with the Norm-Preg group. However, both RUPP and RUPP + Nitrite groups showed similar impairment of the maximal endothelium-dependent Acetylcholine-induced relaxations (Figure 6A). Thus, our results suggest that sodium nitrite effects are, at least in part, dependent on the endothelium.

Hypoxic and ischemic conditions have been suggested as the ideal biochemical environment for reducing ionized nitrite to NO in vivo [15,17,42]. Thus, uteroplacental ischemia caused by the reduction of uterine perfusion pressure induced by RUPP in rats could have this favorable biochemical condition, where nitrite could be reduced to NO, thereby explaining, in part, decreases in maternal systolic blood pressure and prevention of weight loss in fetuses and placentas that were found in the RUPP + Nitrite group (Figure 2A,B and Appendix A). Furthermore, although it has been previously suggested that nitrite reduction is optimized in conditions of hypoxia and ischemia in vivo [43], our results demonstrate that sodium nitrite may also be reduced to NO in physiological conditions (Preg + Nitrite group; Figure 3 and Appendix A). Thus, the present finding could be a reason for concern in pregnancy because hypotensive effects associated with reduced fetal weight were also observed in the Preg + Nitrite group (Figure 1 and Figure 2A and Appendix A), thereby indicating that nitrite-derived NO could have impaired the blood supply in the uteroplacental circulation as a result of intense decreases in maternal systemic blood pressure. Hence, an alternative to minimize the nitrite-derived NO-induced maternal hypotension may be to consider lower doses for inorganic oral treatment or using a dietary organic supplement from beetroot juice [7].

The RUPP-associated reductions in the gelatinolytic activity of vascular MMP-2 (72 kDa and 64 kDa) isoforms were partially restored by sodium nitrite administration in the RUPP + Nitrite group. These results are in accordance with previous reports, which observed that vascular MMP-2 activity was markedly reduced by RUPP [44], and give further support to the physiological role of MMP-2 in the control of vascular function during pregnancy. MMP-2 is a gelatinase known for degrading important components of extracellular matrix, including in the vascular and uterine tissues, and increases in its gelatinolytic activity were previously found in the aorta and uterus of healthy pregnant rats compared with non-pregnant rats [44,45], thereby providing evidence for a critical role of MMP-2 in the normal remodeling of the vascular tissue. Hence, our present results are in agreement with earlier studies that have also found that reductions in the gelatinolytic activity of vascular MMP-2 were associated with increases in collagen content in the aortic segments of RUPP rats [29,32]. Taken together, our present findings suggest that the preeclamptic RUPP model has also induced increases in aortic thickness and stiffness, thereby explaining, in part, the hyper-reactivity to the phenylephrine-induced vascular contractions observed in the RUPP group. Moreover, sodium-nitrite-derived NO could have prevented, at least in part, the development of these vascular abnormalities in aortas from RUPP + Nitrite rats because the gelatinolytic activity of vascular MMP-2 was partially restored in the RUPP + Nitrite group (Figure 7 and Appendix A). Therefore, our present results are in accordance with the notion that vascular tone during healthy pregnancy may be regulated, at least in part, by the delicate balance between gelatinolytic activity of vascular MMP-2 and NO synthesized by endothelium as well as normal hormonal changes of gestation, as previously observed [44]. However, RUPP-induced endothelial dysfunction is followed by decreases in both vascular MMP-2 activity and circulating NO levels with concomitant increases in vascular contraction, but when circulating NO levels were restored by sodium nitrite treatment under the RUPP condition, there was a recovery of the gelatinolytic activity of vascular MMP-2, thus normalizing vascular contraction in aortas from RUPP + Nitrite rats (Figure 5 and Appendix A).

To assess the effects of sodium nitrite treatment on the maternal-fetal interface, NO metabolites were also determined in placental tissue. Interestingly, we found no significant change in the placental NO metabolites of the Preg + Nitrite group compared with the Norm-Preg group (Figure 4 and Appendix A). However, significant and similar higher levels of placental NO metabolites were found in RUPP rats treated (or not) with sodium nitrite (RUPP and RUPP + Nitrite groups) in comparison with the Norm-Preg and Preg + Nitrite groups (Figure 4 and Appendix A). These present findings are supported by previous results in which authors reported that there was an upregulation of inducible NO synthase in the preeclamptic RUPP model and that that was associated with vascular oxidative stress and hypertension in RUPP rats [46]. This previous study may help to explain, in part, our present results regarding increases in placental NO metabolites, thus suggesting that high placental NO levels could be related to the NO synthesized by inducible NO synthase, an enzyme known for causing excessive NO production, but the inducible NO synthase-derived NO is inactivated by superoxide, causing oxidative stress, which may be related to endothelial dysfunction [47]. Importantly, further studies are warranted to examine better the differences between the endothelial NO synthase-derived NO and the inducible NO synthase-derived NO in the placenta because there is evidence that during normal placental development occurs up-regulation of endothelial NO synthase, which seems to be impaired in placental growth restriction of preeclampsia, while inducible NO synthase is down-regulated in the placenta of normal pregnancy but may be up-regulated in the RUPP model of preeclampsia [46,48].

Previous studies have demonstrated that important enzymes in preeclampsia may explain, in part, the results found in the present study, including xanthine oxidoreductase [49] and argininosuccinate synthase 1 [50]. In addition, the end product of xanthine oxidoreductase activity (uric acid) has shown a positive correlation with MMP-2 in long-living individuals [51]. However, future studies should examine if these enzymes may influence the effects of sodium nitrite in pre-eclamptic conditions [52].

Importantly, earlier evidence has shown that the same dose of oral sodium nitrite attenuated the increases in blood pressure of unanesthetized freely-moving male rats [20], thus corroborating the lowering effects on systolic blood pressure observed in anesthetized animals treated with sodium nitrite in the present study. However, previous studies showed no effect on the blood pressure of control and conscious male rats [53] or control and conscious pregnant rats [6], in which animals also received the same dose of oral sodium nitrite. These contrasting results may be explained, in part, by measurements that have been made under anesthesia or by the method of recording the blood pressure (invasive [20] versus non-invasive [6,53] measurements).

Although the present study raises the possibility of protective effects of sodium nitrite in preeclamptic rats, the current results in pregnant rats should be carefully interpreted and extrapolated to other species, particularly humans, because nitrite supplementation may lead to severe adverse events that could potentially be life-threatening in normotensive pregnant women. Furthermore, because we recorded the systolic blood pressure measurements under anesthesia, further studies are needed to confirm our findings in conscious animals. Moreover, because MMP-2 is also important in placental vascular remodeling, further investigations should assess if NO derived from nitrite may affect the activity of MMP-2 in placental tissue from RUPP rats.

## 4. Materials and Methods

### 4.1. Animals and Experimental Protocol

All animal procedures were approved and complied with the guidelines of the Institutional Commission for the Care and Use of Animals (protocol no. 7946200721, 2021) from the Institute of Biosciences of Botucatu, Universidade Estadual Paulista, which are also in accordance with the ARRIVE guidelines.

Female Wistar rats were housed in cages on a 12-h light/dark cycle at 22 ± 2 °C, with ad libitum access to water and food. During the night period, the animals were divided in the ratio of two females to one male rat (Harem system) for mating purposes. The presence of sperm and estrus cells in a vaginal smear was determined on the first gestational day.

On the 14th gestational day, the pregnant rats were divided into the following experimental groups (n = 10 per group): normotensive pregnant rats (Norm-Preg group), pregnant rats treated with sodium nitrite (Preg + Nitrite group), preeclamptic rats (RUPP group), and preeclamptic rats treated with sodium nitrite (RUPP + Nitrite group).

The induction of arterial hypertension by the RUPP method has already been well elucidated to mimic preeclampsia in pregnant rats, as previously detailed [5,23,46]. Briefly, the rats in the RUPP and RUPP + Nitrite groups were anesthetized with Isoflurane (1.5–2%) and underwent a surgical procedure to reduce the blood supply to the uteroplacental circulation. A median incision was performed to implant a silver clip (0.203 mm, internal diameter) in the lower abdominal aorta, above the iliac bifurcation, and two other silver clips (0.100 mm, internal diameter) were implanted in the right and left branches of each ovarian artery on both uterine horns. The RUPP model has been shown to reduce uterine perfusion pressure in the pregnant rat uterus by ~40% [41,52]. The Norm-Preg and Preg + Nitrite groups were also anesthetized with Isoflurane (1.5–2%), but were sham operated on the 14th gestational day with a similar median incision and suturing without a clip implant.

The rats treated with sodium nitrite (Preg + Nitrite and RUPP + Nitrite groups) received a dose of 15 mg/kg/day (equivalent to 0.217 mmol/kg) daily, via gavage, from the 14th gestational day until the end of pregnancy, as previously demonstrated that this dose exerted antihypertensive effects in several experimental models of hypertension [6,36,53,54]. The Norm-Preg and RUPP groups received 0.9% saline solution daily, by gavage, from the 14th gestational day until the end of pregnancy, as schematically represented in the Figure 8.

### 4.2. Maternal Blood Pressure Measurements and Blood, Arteries, and Placenta Harvests

On the 21st gestational day, the females were anesthetized, and a polyethylene catheter (PE50) was inserted into the carotid artery to record systolic blood pressure (SBP) in the anesthetized animals using a data acquisition system (MP150CE, Biopac Systems Inc., Goleta, CA, USA) connected to a computer [23,46]. Subsequently, the animals were euthanized under an overdose of Isoflurane, followed by exsanguination. After euthanasia, a cesarean section was performed, and the fetal and placental weights of each mother were recorded.

Blood samples were collected and centrifuged to obtain plasma, and plasma samples were stored at −80 °C for the analysis.

The thoracic aorta was removed for vascular reactivity experiments, and abdominal aorta segments and placentas were collected and stored at −80 °C for the analysis, as described below.

### 4.3. Determination of NO Metabolites, Nitrite/Nitrate (NOx), in Plasma and Placenta

To determine the NOx, Griess reagent was used, followed by the reduction of nitrous species with vanadium chloride III, as previously described [22,55]. Total NOx values in plasma were expressed in μmol/L, and NOx levels in placenta were expressed in μmol/100 mg of tissue.

### 4.4. Vascular Reactivity Experiments

Segments of the thoracic aorta were dissected and divided into rings with endothelium preserved and rings in which endothelium was mechanically removed. Each ring was suspended between two wire hooks and placed in an organ chamber containing Krebs-Henseleit solution (NaCl 130; KCl 4.7; CaCl_2_ 1.6; KH2PO_4_ 1.2; MgSO_4_ 1.2; NaHCO_3_ 15; glucose 11.1; in mmol/L) kept at 37 °C (pH 7.4) and bubbled with 95% O_2_ and 5% CO_2_, as previously described [22]. The aortic rings were equilibrated in basal tension, and to test the viability of the aorta, KCl-induced maximum contraction was obtained by pipetting KCl (96 mM). Once the KCl-induced maximum contraction was reached, the aortic rings were rinsed with Krebs solution three times, for 15 min each.

Then, aortic rings were stimulated with increasing concentrations of phenylephrine (Phe, 10^−9^ to 10^−4^ M). Once the Phe-induced contraction curve was completed, the aortic rings were rinsed with Krebs solution three times, for 15 min each.

Aortic rings were then pre-contracted with Phe (10^−6^ M), and increasing concentrations (10^−9^ to 10^−4^ M) of acetylcholine (Ach) were added to verify the endothelial function, as previously described [22]. To confirm the involvement of endothelium-derived NO-dependent vasodilation, concentration-response curves for Ach were obtained in the presence of Nω-nitro-L-arginine-methyl ester (L-NAME, 3 × 10^−4^ M) in aortic rings pre-contracted with Phe (10^−6^ M). The ACh-induced relaxation curves were expressed as the % relaxation for the Phe-induced contraction (10^−6^ M), as previously described [22].

### 4.5. Gelatin Zymography for Vascular MMP-2 Activity

The gelatin zymography method was performed in abdominal aortas, as previously described [25,55,56]. For sample preparation, RIPA buffer (1 mM 1,10-ortho-phenanthroline, 1 mM phenylmethanesulfonyl fluoride, and 1 mM N-ethylmaleimide; Sigma-Aldrich, St. Louis, MO, USA) was used, containing protease inhibitor (4-(2-aminoethyl) benzenesulfonyl fluoride (AEBSF), E-64, bestatin, leupeptin, aprotinin, and EDTA) in a proportion of 100 μL RIPA + protease inhibitor for each 10 mg of tissue sample. After homogenization, the protein concentrations of the samples were measured by the Bradford assay (Sigma-Aldrich). Electrophoresis using acrylamide gels (12%) copolymerized with gelatin (0.05%) was used for protein separation. The gels were washed twice, for 30 min at room temperature, in Triton X-100 solution (2%) and incubated for 24 h in a Tris-HCl buffer containing 10 mmol/L of CaCl_2_ at pH 7.4. To stain the gels, Coomassie Brilliant Blue (G-250; Sigma-Aldrich, St. Louis, MO, USA) was used, and to unstain the gels, a methanol solution was used.

Enzyme activity was measured using ImageJ software (1.43u version; NHI, Murfreesboro, TN, USA). Gelatinolytic activity was normalized with an internal standard (2% fetal bovine serum, FBS) to correct for loading and inter-gel variation, and the results were expressed as arbitrary units. The MMP-2 isoforms were identified as bands at 75, 72, and 64 kDa, and the MMP-9 was identified as a band at 92 kDa.

### 4.6. Statistical Analysis

GraphPad Prism^®^ software (version 8.0, San Diego, CA, USA) was used. The results are presented as mean ± SEM, and a probability value (*p*) < 0.05 was considered statistically significant. Shapiro-Wilk tests were applied to verify the normality of the data distribution. Comparisons among groups for values registered on systolic blood pressure, fetal and placental parameters, NO metabolites in plasma and placenta, and gelatinolytic activity were assessed by two-way analysis of variance (ANOVA), with RUPP and sodium nitrite defined as the main effects, followed by Tukey’s test for multiple comparisons. For vascular reactivity experiments, individual concentration-contraction and concentration-relaxation curves were constructed; sigmoidal curves were fitted to the data using the least squares method; the pEC_50_ values were calculated; and comparisons among E_max_ and pEC_50_ values were analyzed. To compare E_max_ and pEC_50_ values, we used one-way ANOVA, and Tukey’s test for post hoc analysis was applied, whereas two-way ANOVA was handled to examine the differences at the phenylephrine and acetylcholine concentrations (point-by-point) among the aortic rings.

## 5. Conclusions

Therefore, the present study provides evidence that the nitrite reduction to NO occurs in RUPP-associated endothelial dysfunction and protects against increases in vascular contraction and systolic blood pressure, and possibly the gelatinolytic activity of vascular MMP-2 is partially restored by sodium nitrite and is involved in the improvement of vascular function. Thus, our findings suggest that the nitrite-NO pathway may be an alternative to provide NO in endothelial dysfunction and vascular hyper-reactivity in preeclampsia.

## Figures and Tables

**Figure 1 ijms-24-12818-f001:**
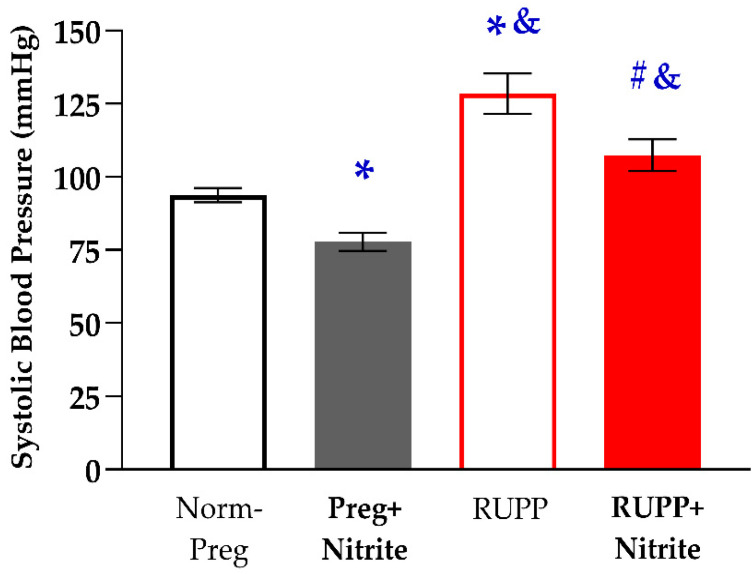
Effects of sodium nitrite on systolic blood pressure (SBP) measured on pregnancy day 21 in the Norm-Preg, Preg + Nitrite, RUPP, and RUPP + Nitrite groups. Values represent the mean ± SEM. * *p* < 0.05 vs. the Norm-Preg group; ^#^
*p* < 0.05 vs. the RUPP group; and ^&^
*p* < 0.05 vs. the Preg + Nitrite group.

**Figure 2 ijms-24-12818-f002:**
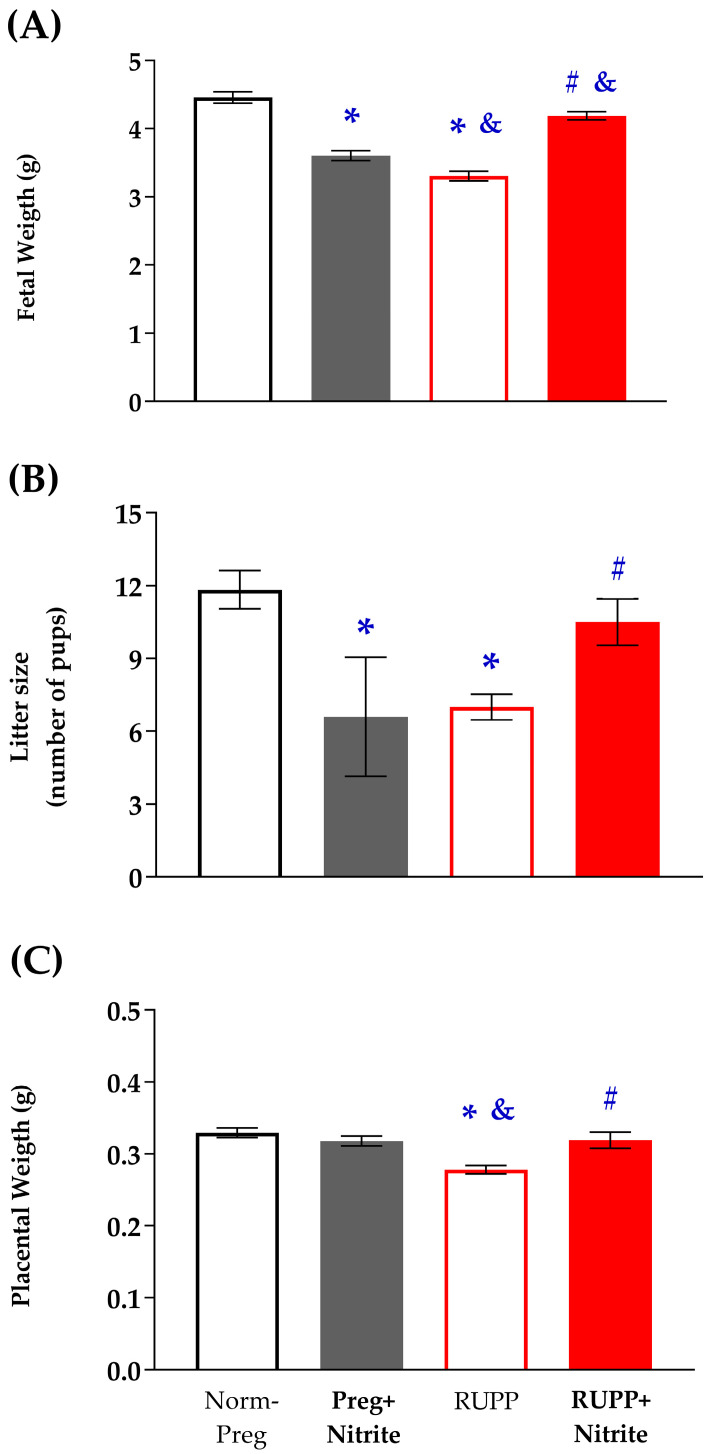
Effects of sodium nitrite on fetal weight (**A**), litter size (number of pups) (**B**), and placental weight (**C**) were recorded in the Norm-Preg, Preg + Nitrite, RUPP, and RUPP + Nitrite groups. Values represent the mean ± SEM. * *p* < 0.05 vs. the Norm-Preg group; ^#^
*p* < 0.05 vs. the RUPP group; and ^&^
*p* < 0.05 vs. the Preg + Nitrite group.

**Figure 3 ijms-24-12818-f003:**
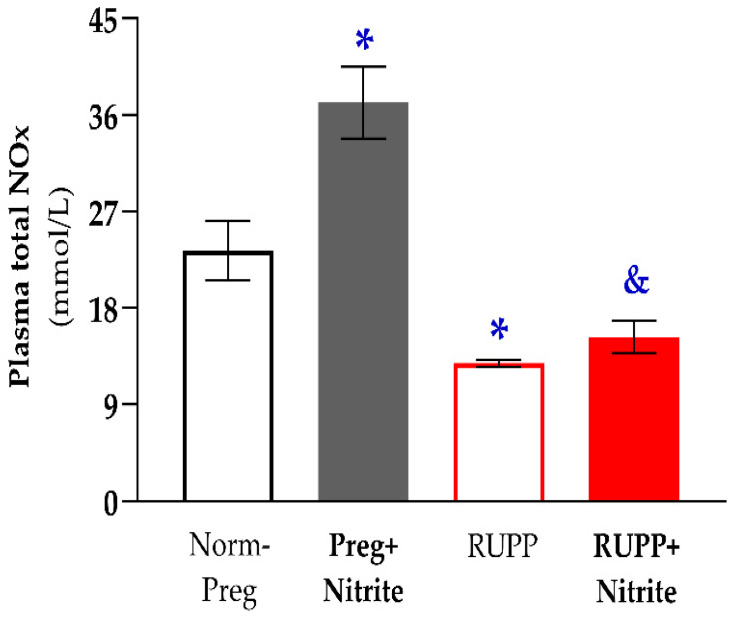
Effects of sodium nitrite on NO metabolites (NOx) in plasma in the Norm-Preg, Preg + Nitrite, RUPP, and RUPP + Nitrite groups. Values represent the mean ± SEM. * *p* < 0.05 vs. the Norm-Preg group, and ^&^
*p* < 0.05 vs. the Preg + Nitrite group.

**Figure 4 ijms-24-12818-f004:**
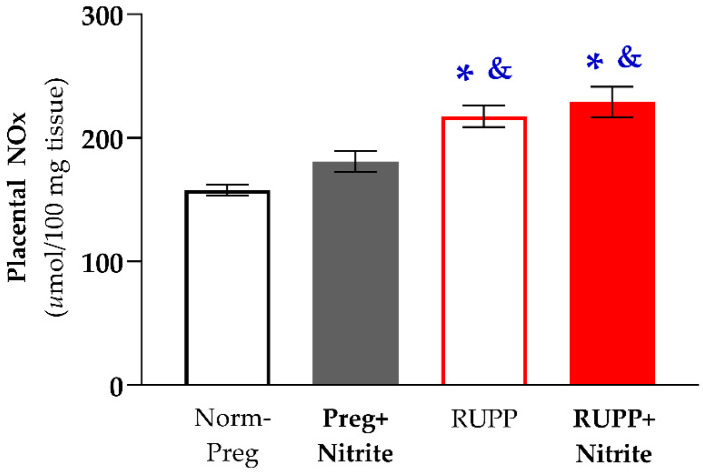
Effects of sodium nitrite on placental NOx levels in the Norm-Preg, Preg + Nitrite, RUPP, and RUPP + Nitrite groups. Values represent the mean ± SEM. * *p* < 0.05 vs. the Norm-Preg group, and ^&^
*p* < 0.05 vs. the Preg + Nitrite group.

**Figure 5 ijms-24-12818-f005:**
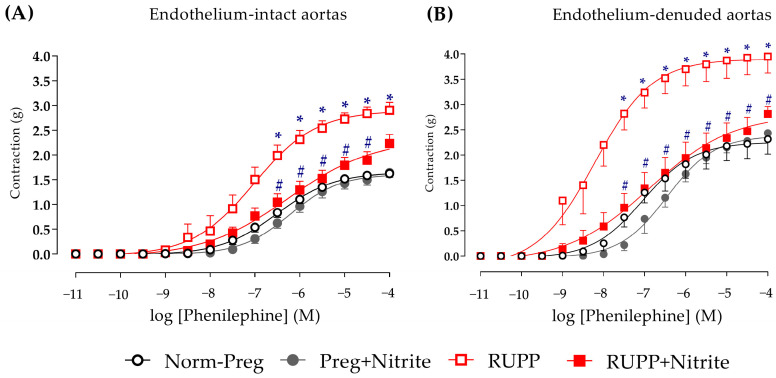
Effects of sodium nitrite on vascular reactivity of the thoracic aorta rings in the contraction induced by phenylephrine with (**A**) or without (**B**) endothelium in the Norm-Preg, Preg + Nitrite, RUPP, and RUPP + Nitrite groups. Values represent the mean ± SEM. * *p* < 0.05 vs. the Norm-Preg group, and ^#^
*p* < 0.05 vs. the RUPP group.

**Figure 6 ijms-24-12818-f006:**
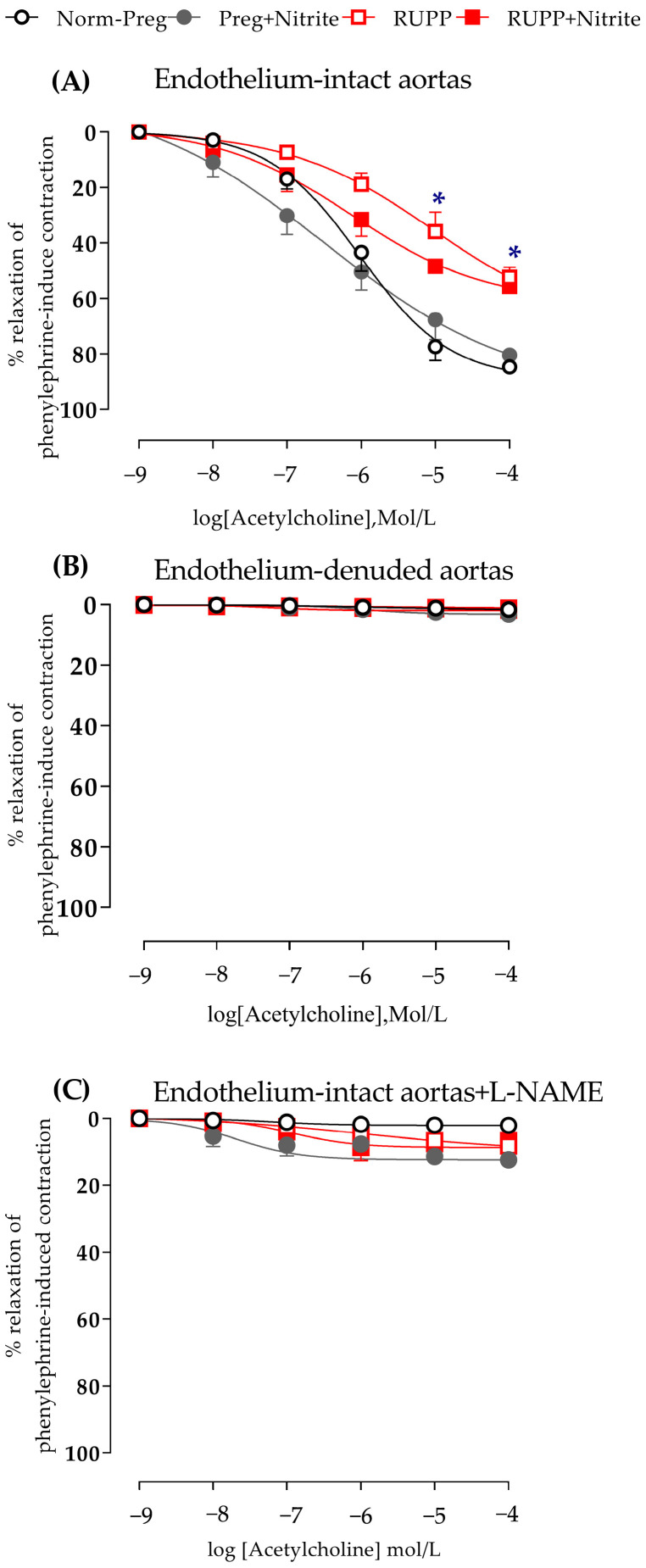
Effects of sodium nitrite on the vascular reactivity of acetylcholine-induced relaxations in the endothelium-intact thoracic aortas (**A**) or endothelium-denuded thoracic aortas (**B**) or endothelium-intact thoracic aortas pre-incubated with L-NAME (**C**) in the Norm-Preg, Preg + Nitrite, RUPP, and RUPP + Nitrite groups. Values represent the mean ± SEM. * *p* < 0.05 for RUPP and RUPP + Nitrite vs. Norm-Preg and Preg + Nitrite groups.

**Figure 7 ijms-24-12818-f007:**
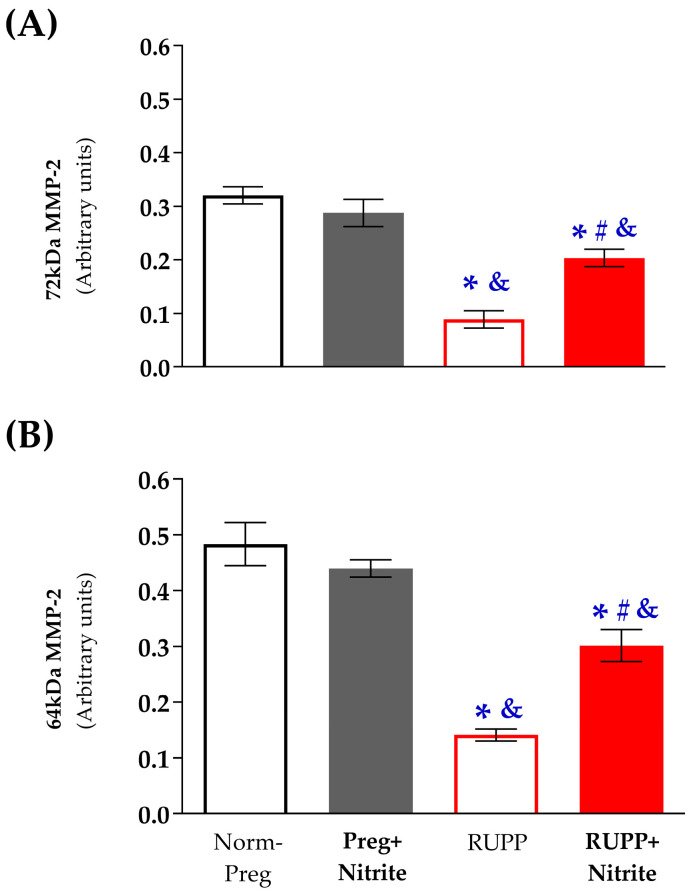
Effects of sodium nitrite on vascular MMP-2 activity. Gelatinolytic activities of 72 kDa MMP-2 (**A**) and 64 kDa MMP-2 (**B**) in the Norm-Preg, Preg + Nitrite, RUPP, and RUPP + Nitrite groups. * *p* < 0.05 vs. the Norm-Preg group; ^#^
*p* < 0.05 vs. the RUPP group; and ^&^
*p* < 0.05 vs. the Preg + Nitrite group.

**Figure 8 ijms-24-12818-f008:**
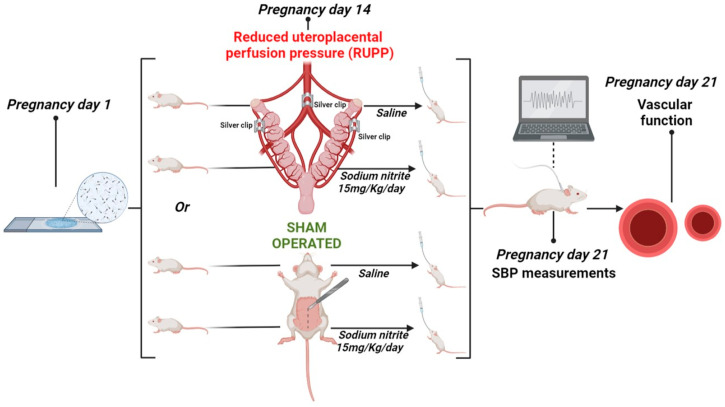
Schematic representation of the RUPP model and experimental protocol. The scheme demonstrates the surgical procedure used to reduce the uteroplacental perfusion pressure and treatment with inorganic sodium nitrite, which reduces NO. This image was created by the BioRender software (https://biorender.com/ (accessed on 20 June 2023), Toronto, ON, Canada).

## Data Availability

The authors declare that all the data supporting the results of the present study are included in the article.

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
