# Peer review of "Sodium Nitrite Attenuates Reduced Activity of Vascular Matrix Metalloproteinase-2 and Vascular Hyper-Reactivity and Increased Systolic Blood Pressure Induced by the Placental Ischemia Model of Preeclampsia in Anesthetized Rats"

_ijms, 2023, doi:10.3390/ijms241612818_

Round 1

Reviewer 1 Report

The aim of the present research is to determine whether oral administration of sodium nitrite prevents hypertension and vascular dysfunction induced by uterine perfusion pressure restriction in an experimental model of preeclampsia in the rat . In addition, this study also examined whether the effects of sodium nitrite in this experimental model of preeclampsia were associated with increases in MMMP-2 activity. It is an interesting study using an experimental model that has been widely used to study many of the pathophysiological features of preeclampsia. However, there are some important issues of the research that need to be clarified.

Title

1.-The word “prevents” in the title should be replaced by “decreases”,” attenuates”, or some other similar word, since the administration of sodium nitrite in the study does not restore metalloproteinase 2 activity levels to those observed in normal pregnant rats.

Materials and Methods

2.-In section 4.2 the authors state that the animals were anesthetized for insertion of a catheter into the carotid artery but it is not clear whether the blood pressure measurement was  performed in the anesthetized animal.

3.-The title of section 4.5 is the same as that of section 4.3. The authors should correct it.

4.- In the statistics section the authors should specify in which of the measured parameters the one way test or the two way test was used.

Results

5.- Are the blood pressure values shown in figure 2 systolic blood pressure readings?. If the animals had these systolic blood pressure values, we could assume that the pressure of normal pregnant rats treated with sodium nitrite would be below the pressure autoregulation mechanism. The authors should clarify whether this is indeed the systolic blood pressure or the measurements are made under anesthesia. In this regard, in a previous paper (Gonçalves-Rizzi et al. Nitric Oxide. 2016 Jul 1;57:71-78. doi: 10.1016/j.niox.2016.05.004. PMID: 27181106), the authors found no effect in normal pregnant rats with the same dose of sodium nitrite. Could it be the decrease in blood pressure observed in the current study influenced by the anesthesia?. Since the decrease in blood pressure is similar in the group of normal rats and the RUPP group treated with sodium nitrite, the authors should demonstrate that this also occurs in conscious animals.

The mean arterial pressure values of conscious pregnant rats with reduced uterine perfusion pressure that have been published in most papers using that experimental model are around 125 mmHg (Spradley FT et al,. Hypertension. 2016 Apr;67(4):740-7. doi: 10.1161/HYPERTENSIONAHA.115.06783; Tam Tam KB et al,. Am J Obstet Gynecol. 2011 Apr;204(4):330.e1-4. doi: 10.1016/j.ajog.2011.01.049, between others). In the current study mean arterial pressures values would be below 125 mmHg. With a systolic blood pressure of around 125 mm Hg the authors cannot say that RUPP rats are hypertensive.  These animals have elevated blood pressure induced by reduced uterine perfusion pressure during gestation but are not hypertensive. Therefore, the authors should discuss the differences with the others studies and replace “hypertensive rats” in the manuscript  by “rats with elevated blood pressure”.

6.- The fetal weight of the offspring depends on the number of fetuses in each animal. Rats with reduced uterine perfusion pressure tend to have fetal resorption, so if they have fewer fetuses they may have a higher weight. In order to compare weights, animals with similar numbers of fetuses should be compared. It is therefore important that the authors, at least, include in the results the mean number of fetuses in each group.

7.- Regarding the results in Figure 7, I do not agree with the authors when they say that sodium nitrite effect are independent of the endothelium. Acetylcholine-induced vasodilatation is more altered in the RUPP group of rats since, as shown in the graph, the relaxation of the vessels is significantly lower than in the control rats with the highest doses of acetylcholine. The administration of sodium nitrite slightly improves this response because in the group of RUPP rats treated with sodium nitrite no differences are observed with respect to the group of normal pregnant rats. This should be changed in the discussion.

8.- Did the authors measure placental activity of MMP-2?. Given the importance of MMP-2 in placental vascular remodeling, it would have been interesting if the authors had determined placental MMP-2 activity in response to sodium nitrite administration.

Discussion

9.-Page 10, line 310-315. The authors state that “no significant change in the placenta NO metabolites of the group treated with sodium nitrite compared with untreated control rats, thus suggesting that nitrite-derived NO could  not have crossed the placental barrier despite.....”. As the authors point out in the discussion the levels of NO metabolites in the placenta do not reflect the bioavailability of NO, so NO metabolites may be increased but that does not indicate that the bioavailability of NO is increased. On the other hand, NO is a gas that diffuses easily so it is hard to believe that NO cannot cross the placenta. It should also be borne in mind that the spiral arterioles flow into the placenta, so that vascular NO can easily reach it. The authors should omit this sentence.

Author Response

Response to Reviewers comments:

We thank the reviewers for helpful comments and suggestions. We have also incorporated all suggested changes, which are highlighted in red in the revised version of the manuscript. Please find below the responses, point by point, to the referees’ comments.

Reviewer 1 #

Comments and Suggestions for Authors

The aim of the present research is to determine whether oral administration of sodium nitrite prevents hypertension and vascular dysfunction induced by uterine perfusion pressure restriction in an experimental model of preeclampsia in the rat. In addition, this study also examined whether the effects of sodium nitrite in this experimental model of preeclampsia were associated with increases in MMMP-2 activity. It is an interesting study using an experimental model that has been widely used to study many of the pathophysiological features of preeclampsia. However, there are some important issues of the research that need to be clarified.

We thank you for your careful revision, helpful comments, and suggestions. Please see below our responses.

Title

1.-The word “prevents” in the title should be replaced by “decreases”,” attenuates”, or some other similar word, since the administration of sodium nitrite in the study does not restore metalloproteinase 2 activity levels to those observed in normal pregnant rats.

We agree with you. Please see our revised version of the title (lines 2-4 of the manuscript) and as follows:

Sodium nitrite attenuates reduced activity of vascular matrix metalloproteinase-2, vascular hyper-reactivity, and increased systolic blood pressure induced by the placental ischemia model of preeclampsia

Materials and Methods

2.-In section 4.2 the authors state that the animals were anesthetized for insertion of a catheter into the carotid artery but it is not clear whether the blood pressure measurement was  performed in the anesthetized animal.

            The blood pressure measurements were recorded in the anesthetized animals. Thereby, we have now included this information in the section 4.2 as follows (lines 380-381 of the manuscript):

4.2. Maternal blood pressure measurements, and blood, arteries and placentas harvests

On the 21st gestational day, the females were anesthetized, and a polyethylene catheter (PE50) was inserted into the carotid artery to record systolic blood pressure (SBP) in the anesthetized animals, using a data acquisition system (MP150CE, Biopac Systems Inc. CA, USA) connected to a computer [23,46]. Subsequently, the animals were euthanized under …

3.-The title of section 4.5 is the same as that of section 4.3. The authors should correct it.

            We thank you for your careful revision. This was an oversight and we have corrected the title of section 4.5 as follows:

4.5 Gelatin zymography for vascular MMP-2 activity

4.- In the statistics section the authors should specify in which of the measured parameters the one way test or the two way test was used.

            We totally agree with you. Please see our revised version of the statistical section (lines 444-454).

4.6. Statistical analysis

GraphPad Prism® software (version 8.0, San Diego, CA, USA) was used. The results are presented as mean ± SEM, and a probability value (P) < 0.05 was considered statistically significant. Shapiro-Wilk tests were applied to verify the normality of data distribution. Comparisons among groups for values registered on systolic blood pressure, fetal and placental parameters, NO metabolites in plasma and placenta, and gelatinolytic activity were assessed by one-way analysis of variance (ANOVA), and Tukey's test for post hoc analysis was applied. For vascular reactivity experiments, individual concentration–contraction and concentration–relaxation curves were constructed; sigmoidal curves were fitted to the data using the least square method, and the pEC50 values were calculated, and comparisons among Emax and pEC50 values were analyzed. To compare Emax and pEC50 values, we used one-way ANOVA, and Tukey's test for post hoc analysis was applied, whereas two-way ANOVA was handled to examine the differences at the phenylephrine and acetylcholine concentrations (point-by-point) among the aortic rings.

Results

5.- Are the blood pressure values shown in figure 2 systolic blood pressure readings?.

            Yes. The values shown in the respective figure are systolic blood pressure values. Please see our revised version of this figure, which now is Figure 1 and as shown in line 97.

If the animals had these systolic blood pressure values, we could assume that the pressure of normal pregnant rats treated with sodium nitrite would be below the pressure autoregulation mechanism.

            Yes. Because systolic blood pressure values were recorded under anesthesia, possibly isoflurane could have systematically affected blood pressure measurements in all experimental groups. However, although the measurements could be below the pressure autoregulation mechanism, our present results demonstrate that RUPP elevated the systolic blood pressure, and sodium nitrite consistently reduced the systolic blood pressure in the animals submitted (or not) to the RUPP model. Please find below an explanation and reason by which we needed to record systolic blood pressure under anesthesia.

The authors should clarify whether this is indeed the systolic blood pressure or the measurements are made under anesthesia.

The values shown in figure 1 are indeed the systolic blood pressure and the measurements were made under anesthesia. Please see our revised version in which changes are now highlighted in red color in the manuscript (lines, 97, 380, and 381).

In this regard, in a previous paper (Gonçalves-Rizzi et al. Nitric Oxide. 2016 Jul 1;57:71-78. doi: 10.1016/j.niox.2016.05.004. PMID: 27181106), the authors found no effect in normal pregnant rats with the same dose of sodium nitrite. Could it be the decrease in blood pressure observed in the current study influenced by the anesthesia?.

No. Although the decreases in systolic blood pressure observed in the current study were possibly influenced by the anesthesia in all experimental groups, in our previous article the systolic blood pressure measurements were recorded in conscious animals using a tail-cuff plethysmography method. However, the tail‐cuff technique requires handling and restraint of the animal during measurements, which may cause stress and thus affecting the blood pressure and undermining reliability of the results, as previously demonstrated (J Am Heart Assoc. 2017 Jun; 6(6): e005204, doi: 10.1161/JAHA.116.005204).

Importantly, the RUPP model restricts blood flow to the hindquarters, thus it is not suitable to record the increases in arterial blood pressure measured via tail cuff plethysmography, as previously suggested (Sci Rep 2019 Jul 2;9(1):9565. doi: 10.1038/s41598-019-45959-6). However, although the present systolic blood pressure measurements could be below the pressure autoregulation mechanism, all groups were submitted to anesthesia, and thereby it is possible that all measurements could be underestimated. But the present results demonstrate that RUUP elevated the systolic blood pressure, while treatment with sodium nitrite reduced it in animals submitted (or not) to the RUPP model, thus suggesting that preeclamptic model increases the systolic blood pressure and that nitrite was reduced to NO, and NO decreases systolic blood pressure.

Therefore, we have also included this statement at the last paragraph of Discussion. Please see our revised version in which changes are now highlighted in red color in the manuscript (lines 338-340).

“Furthermore, because we have recorded the systolic blood pressure measurements under anesthesia, future studies need to confirm our findings in conscious animals.”

Since the decrease in blood pressure is similar in the group of normal rats and the RUPP group treated with sodium nitrite, the authors should demonstrate that this also occurs in conscious animals.

            We agree with you that it would be interesting to demonstrate that the decreases in systolic blood pressure also occur in conscious animals treated with sodium nitrite. However, we have performed a pilot study in which we have tried several times to exteriorize the catheter at the back of the neck of the animals, but we almost always lost the measurements either due to clot embolism into the cannula or disconnection of the cannula from the transducer. Thereby, the possible alternative for us to invasively and directly measure the blood pressure was by recording it in the anesthetized animals. Thus, future studies need to confirm our findings with telemetry methods that record blood pressure online in conscious animals, being considered the “gold‐standard” technique.

Please see our revised version of the manuscript in which this limitation was included in the last paragraph of the Discussion, with the respective changes highlighted in red color (lines

Please see our revised version in which changes are now highlighted in red color in the manuscript (lines 338-340) an as follows:

“Furthermore, because we have recorded the systolic blood pressure measurements under anesthesia, future studies need to confirm our findings in conscious animals.”

The mean arterial pressure values of conscious pregnant rats with reduced uterine perfusion pressure that have been published in most papers using that experimental model are around 125 mmHg (Spradley FT et al,. Hypertension. 2016 Apr;67(4):740-7. doi: 10.1161/HYPERTENSIONAHA.115.06783; Tam Tam KB et al,. Am J Obstet Gynecol. 2011 Apr;204(4):330.e1-4. doi: 10.1016/j.ajog.2011.01.049, between others). In the current study mean arterial pressures values would be below 125 mmHg. With a systolic blood pressure of around 125 mm Hg the authors cannot say that RUPP rats are hypertensive.  These animals have elevated blood pressure induced by reduced uterine perfusion pressure during gestation but are not hypertensive. Therefore, the authors should discuss the differences with the others studies and replace “hypertensive rats” in the manuscript  by “rats with elevated blood pressure”.

            We totally agree with you. Indeed, the systolic blood pressure measurements in the present study could be below the pressure autoregulation mechanism since it was recorded in animals under anesthesia. Therefore, we have now replaced hypertensive rats by the RUPP rats with elevated systolic blood pressure in the revised version of the manuscript. Thus, we have also stated that anesthesia could have underestimated the systolic blood pressure measurements, and thereby explaining in part the differences with previous studies. Please see our revised version of the manuscript in which this limitation was included in Results (line 97) and Discussion (line 209) and respective changes are now highlighted in red color,

6.- The fetal weight of the offspring depends on the number of fetuses in each animal. Rats with reduced uterine perfusion pressure tend to have fetal resorption, so if they have fewer fetuses they may have a higher weight. In order to compare weights, animals with similar numbers of fetuses should be compared. It is therefore important that the authors, at least, include in the results the mean number of fetuses in each group.

            We agree with you that it is important for the readers to know the number of fetuses in the manuscript. Thus, we have now included in the results the average of the number of fetuses in each group (line 125). Please, see our revised version of this figure. However, we respectfully disagree with you regarding the RUPP rats having fewer number of fetuses and with concomitant higher fetal weight. In this regard, we and others have observed that both total number of fetuses and fetal weights were decreased in RUPP rats, as previously found (Biochem Pharmacol 2017 Dec 15;146:101-116, doi: 10.1016/j.bcp.2017.09.005 ; Am J Physiol Regul Integr Comp Physiol 2016 Sep 1;311(3):R505-21, doi: 10.1152/ajpregu.00137.2016).

7.- Regarding the results in Figure 7, I do not agree with the authors when they say that sodium nitrite effect are independent of the endothelium. Acetylcholine-induced vasodilatation is more altered in the RUPP group of rats since, as shown in the graph, the relaxation of the vessels is significantly lower than in the control rats with the highest doses of acetylcholine. The administration of sodium nitrite slightly improves this response because in the group of RUPP rats treated with sodium nitrite no differences are observed with respect to the group of normal pregnant rats. This should be changed in the discussion.

            We agree with you, and we have now changed this statement in the Discussion as well. To clarify to the readers, we have included in the text that RUPP+Nitrite rats showed trends and a slight improvement on Acetylcholine-induced relaxations at the concentration of 10-6 mol/L, thus, our results suggest that sodium nitrite effects are, at least in part, dependent on endothelium. In addition, to clarify to the readers, we have also revised the legend of the figure 6, and now it is stating that the *P < 0.05 for RUPP and RUPP+Nitrite vs. Norm-Preg and Preg+Nitrite groups on the respective concentrations that were statistically different.

Please see our revised version below in which these statements were changed in Discussion and respective changes are now highlighted in red color in the revised version of the manuscript.

  1. Discussion (lines 208 and 214).

The present findings demonstrate, for the first time, that sodium nitrite attenuates the increases in systolic blood pressure and counteracts the vascular hyper-reactivity, and prevents fetal and placental weight loss induced by RUPP model of preeclampsia. This study also shows that sodium nitrite partially restores the gelatinolytic activity of vascular MMP-2 impaired by RUPP. Thus, our results suggest that reduction of nitrite to NO occurs during placental ischemia and endothelial dysfunction, and thereby protecting against RUPP-associated increases in vascular contraction and in systolic blood pressure.

Discussion (lines 255-261)

“… The present study demonstrates that endothelial dysfunction featured by impaired acetylcholine-induced vasodilation was slightly improved in the RUPP+Nitrite group, because no differences were found with the respective concentration-induced relaxations of the Acetylcholine (10-6 mol/L, Figure 6A) in comparison with Norm-Preg group. However, both RUPP and RUPP+Nitrite groups showed similar impairment of the maximal endothelium-dependent Acetylcholine -induced relaxations (Figure 6A). Thus, our results suggest that sodium nitrite effects are, at least in part, dependent on endothelium.”

8.- Did the authors measure placental activity of MMP-2?. Given the importance of MMP-2 in placental vascular remodeling, it would have been interesting if the authors had determined placental MMP-2 activity in response to sodium nitrite administration.

            We did not measure placental activity of MMP-2 because our present results related to the NO metabolites in placental homogenates demonstrate that RUPP may induce a exaggerated increase in placental NO levels due to a complex upregulation of inducible NO synthase activity, as previously suggested (J Cell Mol Med 2013, 17, 1300–1307, doi:10.1111/jcmm.12106) and that treatment with sodium nitrite showed no statistical differences related to the placental NO metabolites in the Preg+Nitrite compared with Norm-Preg group as well as RUPP compared with RUPP+Nitrite group. Thus, the eventual reduction of nitrite to NO in the placenta tissue may be irrelevant in front of large amount of NO derived from inducible NO synthase, which is activated by RUPP model (J Cell Mol Med 2013, 17, 1300–1307, doi:10.1111/jcmm.12106). Therefore, further studies should investigate if NO derived from nitrite may affect the activity of MMP-2 in placental tissue from RUPP rats.

Please see our revised version in which this limitation was included in the last paragraph of the Discussion, and respective changes are now highlighted in red color in the manuscript (lines 340-342) and as follows:

“Moreover, because MMP-2 is also important in the placental vascular remodeling, further investigations should assess if NO derived from nitrite may affect the activity of MMP-2 in placental tissue from RUPP rats.”

Discussion

9.-Page 10, line 310-315. The authors state that “no significant change in the placenta NO metabolites of the group treated with sodium nitrite compared with untreated control rats, thus suggesting that nitrite-derived NO could not have crossed the placental barrier despite.....”. As the authors point out in the discussion the levels of NO metabolites in the placenta do not reflect the bioavailability of NO, so NO metabolites may be increased but that does not indicate that the bioavailability of NO is increased. On the other hand, NO is a gas that diffuses easily so it is hard to believe that NO cannot cross the placenta. It should also be borne in mind that the spiral arterioles flow into the placenta, so that vascular NO can easily reach it. The authors should omit this sentence.

            We totally agree with you, and thereby we have now revised this statement in the Discussion (lines 307-310) and as follows:

“To assess effects of sodium nitrite treatment on maternal-fetal interface, NO metabolites were also determined in placental tissue. Interestingly, we found no significant change in the placental NO metabolites of the Preg+Nitrite group compared with the Norm-Preg group (Figure 4 and Supplementary table S1). However, significant and similar higher levels of placental NO metabolites were found in RUPP rats treated (or not) with sodium nitrite (RUPP and RUPP+Nitrite groups) in comparison with Norm-Preg and Preg+Nitrite groups (Figure 4 and Supplementary table S1). These present findings are supported by previous results in which authors reported that there was an upregulation of inducible NO synthase in the preeclamptic RUPP model, and that was associated with vascular oxidative stress …”

Reviewer 2 Report

The experimental work of Zanetoni Martins et al. is an animal study on the role of sodium nitrite supplementation in the RUPP model of preeclampsia. The paper is of merit and experimental design is appropriate, however I have several comments:

- In the Introduction section Authors stated: "Importantly, trophoblast differentiation, motility, and invasion may all be regulated by NO, thus dysregulation of endogenous NO production in the uteroplacental circulation may contribute to pregnancy complications in preeclampsia". However, trophoblast function is essential for correct placentation but has not been demonstrated to play a direct role in pregnancy complications in preeclampsia. Apart from placentation, NO dysfunction implies a systemic endothelial dysfunction that contributes to adverse pregnancy outcomes including preeclampsia.

- In the results section the title of paragraphs should be shorter and summarise the experimental approach e.g. sodium nitrite supplementation and serum NO metabolites instead of "Sodium nitrite protects against significant reductions in NO metabolites levels of 125 maternal circulation without affecting the increases in placental NO metabolites induced 126 by RUPP"

- Authors stated: "...suggesting that sodium nitrite-derived NO underlies antihypertensive effect 175 and occurs independently of endothelial dysfunction caused by uteroplacental ischemia".  However, this conclusion is not supported by their results and it represents a speculation that should be mention in the discussion and not in the results session.

- In all the histograms of the Figures I would simplify symbols for statistical significance e.g. * (RUPP vs. Norm preg).

- Figure 8A is useless since one can barely see the bands.

- Authors should mention that sodium nitrite supplementation may lead to severe adverse events potentially life treating in humans

- Important players in NO homeostasis that may partially explain the results found by Authors are xanthine oxidoreductase (doi: 10.1016/j.niox.2013.02.081) and arginino succinate synthase 1 (doi: 10.1007/s13577-023-00901-x), both showing emerging roles in preeclampsia. Furthermore in humans uric acid, the end product of xanthine oxidoreductase activity, has shown to positively associate with MMP-2 (doi: 10.1155/2020/8635158). 

Several sentences are difficult to follow and must be improved, e.g. "Accordingly, although decreases in the ge- 81 latinolytic activity of vascular MMP-2 [23,25,29] associated with reductions in the metab- 82 olites of NO in the maternal circulation [5,30] with subsequent development of hyperten- 83 sion in preeclamptic pregnancies have also been observed, no previous study has not yet......"

I suggest Authors to use short sentences (max 2 lines) and overall to divide the content to more than one sentence. 

Author Response

Response to Reviewers comments:

We thank the reviewers for helpful comments and suggestions. We have also incorporated all suggested changes, which are highlighted in red in the revised version of the manuscript. Please find below the responses, point by point, to the referees’ comments.

Reviewer 2 #

Comments and Suggestions for Authors

The experimental work of Zanetoni Martins et al. is an animal study on the role of sodium nitrite supplementation in the RUPP model of preeclampsia. The paper is of merit and experimental design is appropriate, however I have several comments:

We thank you for your careful revision, helpful comments, and suggestions. Please see below our responses.

- In the Introduction section Authors stated: "Importantly, trophoblast differentiation, motility, and invasion may all be regulated by NO, thus dysregulation of endogenous NO production in the uteroplacental circulation may contribute to pregnancy complications in preeclampsia". However, trophoblast function is essential for correct placentation but has not been demonstrated to play a direct role in pregnancy complications in preeclampsia. Apart from placentation, NO dysfunction implies a systemic endothelial dysfunction that contributes to adverse pregnancy outcomes including preeclampsia.

We agree with you. Although there is previous evidence suggesting that NO dysfunction may be involved in deficient trophoblast invasion, further studies need to investigate if endogenous NO is essential for physiological placentation. Therefore, we have now revised this statement in the Introduction (lines 45-51) and as follows:

“NO is a vasodilator molecule produced by vascular endothelial cells, controls maternal blood pressure in pregnancy, and regulates uteroplacental blood flow [6,7]. Placental development and maturation are complex processes that require coordinated regulation of trophoblast invasion, differentiation, and proliferation in the maternal decidua [8]. Importantly, trophoblast differentiation, motility, and invasion may all be dysregulated in preeclampsia [9]. Moreover, dysfunction of the endogenous NO production in the maternal circulation may also contribute to pregnancy complications including preeclampsia [9]….”

- In the results section the title of paragraphs should be shorter and summarise the experimental approach e.g. sodium nitrite supplementation and serum NO metabolites instead of "Sodium nitrite protects against significant reductions in NO metabolites levels of 125 maternal circulation without affecting the increases in placental NO metabolites induced 126 by RUPP"

We totally agree with you. Please see our revised titles of paragraphs in the Results section as follow:

2.1. Sodium nitrite effects on systolic blood pressure and fetal and placental parameters

2.2. Sodium nitrite treatment and circulating and placental levels of NO metabolites

2.3. Sodium nitrite effects on vascular hyperreactivity and endothelial dysfunction

2.4. Sodium nitrite effects on gelatinolytic activity of vascular MMP-2

- Authors stated: "...suggesting that sodium nitrite-derived NO underlies antihypertensive effect 175 and occurs independently of endothelial dysfunction caused by uteroplacental ischemia".  However, this conclusion is not supported by their results and it represents a speculation that should be mention in the discussion and not in the results session.

We totally agree with you, and therefore, we have now removed this statement from the Results section, and we have mentioned it in the Discussion section. Please see our revised version of the manuscript with changes highlighted in red color.

Results section (lines 171-174):

“Acetylcholine-induced vasodilation in aortic rings with intact-endothelium was impaired in both RUPP and RUPP+Nitrite groups in comparison with untreated and treated sham-operated rats (Norm-Preg and Preg+Nitrite groups, Figure 6A and Supplementary table S1).”

Discussion section (lines 255-261):

The present study demonstrates that endothelial dysfunction featured by impaired acetylcholine-induced vasodilation was slightly improved in the RUPP+Nitrite group because no differences were found in the respective concentration-induced relaxations of the Acetylcholine (10-6 mol/L, Figure 6A) in comparison with Norm-Preg group. However, both …”

- In all the histograms of the Figures I would simplify symbols for statistical significance e.g. * (RUPP vs. Norm preg).

We respectfully disagree with you, because important differences were found in the Preg+Nitrite group when compared to the Norm-Preg group, in particular regarding systolic blood pressure, fetal weight, litter size, and NO metabolites in plasma. Therefore, we have also included this information in the limitations of the present study at the last paragraph of discussion (lines 334-338) as follows.

“Although the present study raises the possibility of protective effects of sodium nitrite in preeclamptic rats, the current results in pregnant rats should be carefully interpreted and extrapolated to other species, particularly in relation to human, because nitrite supplementation may lead to severe adverse events potentially life treating in normotensive pregnant women.”

- Figure 8A is useless since one can barely see the bands.

            We agree with your suggestion. We have now removed this panel from figure 7, and the readers can see all zymography gels that are shown in supplementary material. Please see this revised statement as follows:

Lines 188-191):

2.4. Sodium nitrite effects on gelatinolytic activity of vascular MMP-2

Zymography analysis of the abdominal aorta tissue homogenates revealed proteolytic bands corresponding to the 72 kDa and 64 kDa MMP-2 isoforms …”

The legend of the respective figure was also revised as follows:

Figure 7. Effects of sodium nitrite on vascular MMP-2 activity. Gelatinolytic activities of 72 kDa MMP-2 (A) and 64 kDa MMP-2 (B) in the Norm-Preg, Preg+Nitrite, RUPP, and RUPP+Nitrite groups. *P < 0.05 vs Norm-Preg group, #P < 0.05 vs RUPP group, and &P < 0.05 vs Preg+Nitrite group.

- Authors should mention that sodium nitrite supplementation may lead to severe adverse events potentially life treating in humans

            We thank you for your careful revision, and thereby we have now mentioned this statement in the last paragraph of the Discussion section (lines 334-338) and as follows:

“Although the present study raises the possibility of protective effects of sodium nitrite in preeclamptic rats, the current results in pregnant rats should be carefully interpreted and extrapolated to other species, particularly in relation to human, because nitrite supplementation may lead to severe adverse events potentially life treating in normotensive pregnant women…”

- Important players in NO homeostasis that may partially explain the results found by Authors are xanthine oxidoreductase (doi: 10.1016/j.niox.2013.02.081) and arginino succinate synthase 1 (doi: 10.1007/s13577-023-00901-x), both showing emerging roles in preeclampsia. Furthermore in humans uric acid, the end product of xanthine oxidoreductase activity, has shown to positively associate with MMP-2 (doi: 10.1155/2020/8635158).

            We agree with you and thank you for your suggestions. Therefore, we have added the following statements at the end of the Discussion section (lines 328-333) and as follows:.

“Previous studies have demonstrated that important enzymes in preeclampsia may explain, in part, the results found in the present study including xanthine oxidoreductase [49] (doi: 10.1016/j.niox.2013.02.081)  and argininosuccinate synthase 1 [50] (doi: 10.1007/s13577-023-00901-x). In addition, the end product of xanthine oxidoreductase activity (uric acid) has shown positive correlation with MMP-2 in long-living individuals [51] (doi: 10.1155/2020/8635158). However, future studies should examine if these enzymes may influence the effects of sodium nitrite in pre-eclamptic conditions.”

Comments on the Quality of English Language

Several sentences are difficult to follow and must be improved, e.g. "Accordingly, although decreases in the ge- 81 latinolytic activity of vascular MMP-2 [23,25,29] associated with reductions in the metab- 82 olites of NO in the maternal circulation [5,30] with subsequent development of hyperten- 83 sion in preeclamptic pregnancies have also been observed, no previous study has not yet......"

I suggest Authors to use short sentences (max 2 lines) and overall to divide the content to more than one sentence.

            We agree with you. Please see this sentence revised below. In addition, we thank you for your careful revision, and we have also revised the whole manuscript as per your suggestions, and changes are highlighted in red color (lines 82-85) in the revised version of the whole manuscript and as follows:

“Accordingly, decreases in the gelatinolytic activity of vascular MMP-2 [23,25,29] associated with reduced NO metabolites in the maternal circulation [5,30] and development of hypertension in preeclamptic pregnancies have also been observed. However, no previous study has not yet…”

Round 2

Reviewer 1 Report

Thank you for the responses. The authors have significantly improved the manuscript by heeding the reviewer's recommendations. However, there are still some aspects that should be considered.

First, it would be more appropriate to apply a two-way ANOVA to the measured parameters. Although I think that the statistical significance would not be very different, this would allow corroboration of the results and conclusions made by the authors in the manuscript.

And second, I understand the authors' explanations as to why the same dose of sodium nitrite used in the present study has no effect on conscious pregnant rats. However, this does not demonstrate that the effect of sodium nitrite in RUPP rats is similar in conscious animals. Furthermore the authors include in the manuscript references from several authors in which this dose of sodium nitrite has been used in SD and Wistar control rats and in none of them is any effect on blood pressure observed. Therefore this aspect should be discussed a little more in the manuscript and should be included in the title. It should be made clear that this effect is exclusive to anesthetized animals, until proven otherwise.

Author Response

Reviewer 1 #

Comments and Suggestions for Authors

Thank you for the responses. The authors have significantly improved the manuscript by heeding the reviewer's recommendations. However, there are still some aspects that should be considered.

We thank you again for helpful comments and suggestions. Now, we have incorporated all your suggested changes, which are highlighted in red in the revised version (R2) of the manuscript. Please find below the responses.

First, it would be more appropriate to apply a two-way ANOVA to the measured parameters. Although I think that the statistical significance would not be very different, this would allow corroboration of the results and conclusions made by the authors in the manuscript.

We totally agree with you. Thus, all measured parameters were submitted to a two-way ANOVA, with RUPP and sodium nitrite defined as the main effects, followed by Tukey's test for multiple comparisons. Please see changes highlighted in red in the revised version (R2) of the manuscript (lines 454-457) and as follows:

“Comparisons among groups for values registered on systolic blood pressure, fetal and placental parameters, NO metabolites in plasma and placenta, and gelatinolytic activity were assessed by two-way analysis of variance (ANOVA), with RUPP and sodium nitrite defined as the main effects, followed by Tukey's test for multiple comparisons.”

And second, I understand the authors' explanations as to why the same dose of sodium nitrite used in the present study has no effect on conscious pregnant rats. However, this does not demonstrate that the effect of sodium nitrite in RUPP rats is similar in conscious animals.

                We agree with you. Thus, we have now included this information as limitations of the present study (lines 347-349) and as follows:

“Furthermore, because we have recorded the systolic blood pressure measurements under anesthesia, further studies need to confirm our findings in conscious animals.”

Furthermore the authors include in the manuscript references from several authors in which this dose of sodium nitrite has been used in SD and Wistar control rats and in none of them is any effect on blood pressure observed. Therefore this aspect should be discussed a little more in the manuscript and should be included in the title. It should be made clear that this effect is exclusive to anesthetized animals, until proven otherwise.

                We agree with you. Therefore, we have now discussed this aspect in the Discussion section (lines 334-342). Also, we have now included the following words in the title of the revised manuscript as well (line 5). Please see the respective changes highlighted in red in the revised version (R2) of the manuscript as follows:

                Discussion section (lines 334-342):

Importantly, earlier evidence has shown that the same dose of oral sodium nitrite attenuated the increases in blood pressure of unanesthetized freely-moving male rats [20], thus corroborating the lowering effects on systolic blood pressure observed in anesthetized animals treated with sodium nitrite in the present study. However, previous studies showed no effect on the blood pressure of control and conscious male rats [53] and in control and conscious pregnant rats [6], in which animals also received the same dose of oral sodium nitrite. These contrasting results may be explained, in part, by measurements that have been made under anesthesia or by the method of recording the blood pressure (invasive [20] versus non-invasive [6, 53] measurements).

New version of the title (line 5):

Sodium nitrite attenuates reduced activity of vascular matrix metalloproteinase-2, vascular hyper-reactivity, and increased systolic blood pressure induced by the placental ischemia model of preeclampsia in anesthetized rats

Reviewer 2 Report

The Authors revised the manuscript addressing all my concerns. The paper now is suitable for publication.

Author Response

The Authors revised the manuscript addressing all my concerns. The paper now is suitable for publication.

We thank you for your careful revision. Please find the revised version of the manuscript (R2) with changes highlighted in red
